# Rapid Molecular Diagnostics in Vulvovaginal Candidosis

**DOI:** 10.3390/diagnostics14202313

**Published:** 2024-10-17

**Authors:** Karolina Akinosoglou, Georgios Schinas, Despoina Papageorgiou, Eleni Polyzou, Zoe Massie, Sabriye Ozcelik, Francesca Donders, Gilbert Donders

**Affiliations:** 1School of Medicine, University of Patras, 26504 Rio, Greece; akin@upatras.gr (K.A.); georg.schinas@gmail.com (G.S.); dspn.pap96@gmail.com (D.P.); polyzou.el@gmail.com (E.P.); 2Department of Internal Medicine and Infectious Diseases, University General Hospital of Patras, 26504 Rio, Greece; 3Femicare, Clinical Research for Women, 3300 Tienen, Belgium; zoe.massie@femicare.net (Z.M.); sabriye.ozcelik@femicare.net (S.O.); francesca.donders@gmail.com (F.D.); 4Regional Hospital Heilig Hart, 3300 Tienen, Belgium; 5Department of Obstetrics and Gynecology, University Hospital Antwerpen, 2650 Antwerp, Belgium

**Keywords:** vulvovaginal candidiasis, vulvovaginal candidosis, diagnostic techniques and procedures, polymerase chain reaction, DNA probes, molecular diagnostic techniques, nucleic acid amplification techniques

## Abstract

Background/Objectives: Vulvovaginal candidosis (VVC) is a common condition among women, with current diagnostic methods relying on clinical evaluation and laboratory testing. These traditional methods are often limited by the need for specialized training, variable performance, and lengthy diagnostic processes, leading to delayed treatment and inappropriate antifungal use. This review evaluates the efficacy of molecular diagnostic tools for VVC and provides guidance on their application in clinical practice. Methods: A literature search was conducted using PubMed to identify studies evaluating rapid diagnostic tests specifically for vulvovaginal *Candida* isolates. Inclusion criteria focused on studies utilizing molecular diagnostics for the detection of *Candida* species in VVC. Articles discussing non-vaginal *Candida* infections, non-English studies, and animal or in vitro research were excluded. Results: Twenty-three studies met the inclusion criteria, predominantly evaluating nucleid acid amplification tests/polymerase chain reaction (NAAT/PCR) assays and DNA probes. PCR/NAAT assays demonstrated high sensitivity and specificity (>86%) for VVC diagnosis, outperforming conventional diagnostic methods. Comparatively, DNA probes, while simpler, exhibited lower sensitivity. The included studies were mostly observational, with only one randomized controlled trial. Emerging diagnostic technologies, including artificial intelligence and integrated testing models, show promise for improving diagnostic precision and clinical outcomes. Conclusions: Molecular diagnostics offer a significant improvement in VVC management, though traditional methods remain valuable in resource-limited settings.

## 1. Introduction

Vulvovaginal candidosis (VVC) remains a significant health concern among women. It is estimated that 75% of women of reproductive age will have at least one episode of VVC. Approximately 7% of European women will have recurrent infections, and 9% of women between 25 and 34 years of age [1,2]. The most common symptoms include vulvar itching, pain, sometimes also dysuria or dyspareunia, and abnormal vaginal discharge. The feeling of discomfort in combination with a lack of severe physical disability, a general lack of readily available and properly trained physicians, in the current era of readily available internet information, discourages women from seeking medical advice, rather often leading to inappropriate over-the-counter treatments [3].

Compared to the microbiological diagnosis, the clinical diagnosis exhibits a sensitivity of 70.3% and a specificity of 83.7% [4]. The collection of a vaginal swab for testing, pH assessment, an odor test, and wet mount microscopy is recommended by the current guidelines for the management of abnormal vaginal discharge [5,6]. According to these guidelines, VVC should be suspected in the presence of a normal vaginal pH (<4.5) together with the clinical symptoms [7], but we disagree with this, as pH can vary from low to very high in the presence of *Candida* [8,9]. Microscopic examination of the vagina with saline or 10% KOH using light or phase-contrast microscopy is a very easy, rapid tool to confirm *Candida* infection [10,11,12]. Furthermore, the skills to master fresh wet mount microscopy can be mastered to an excellent level in a short training time [11] of only 10 h. When budding yeast or hyphae/pseudohyphae are noted, then a diagnosis of VVC can be confirmed [13,14]. Nonetheless, sensitivity can be very low in the case that too few microorganisms are present [10].

A lack of willingness to perform respective tests or proper training in the preparation and interpretation of a wet mount may also result in insufficient clinical evaluation [15]. A recent study has shown that, among women with respective symptoms, microscopy was conducted in only 17.4% of patients [16]. Microscopic assessment of vaginal discharge was not conducted in 37% of 150 clinic visits, and 42% of 50 physicians did not use microscopy in their evaluation of vulvovaginal disease, while, in over 90% of office visits, even the pH measurement of vaginal discharge was omitted [17]. A review of 149,934 American patients’ records indicated that over 60% lacked procedure codes for any type of vaginitis diagnostic testing [18]. In the Netherlands, only 16% (61/380) of general practitioners reported “always” or “often” using microscopy to diagnose VVC, and merely 7.9% (30/380) reported “always” or “often” using culture for this purpose [19]. Other studies identified high misdiagnosis rates of bacterial vaginosis (BV) and VVC, regardless of the use of microscopy, suggesting that inadequate use of microscopy might be a contributing factor [20]. The underutilization of these straightforward in-clinic tests often results in inadequate treatment, with up to 47% of patients receiving one or more inappropriate prescriptions, and 54% of visits involving treatment without sufficient evaluation [16,17,21], indicating that appropriate treatment occurs in less than half of the cases, contributing to the growing problem of antifungal resistance mirrored in increasing azole refractory symptoms and recurrent disease (rVVC) [4,20]. According to a recent survey conducted to measure awareness of vaginitis clinical guidelines and use of in-office diagnostic tools, physicians demonstrated limited awareness of the recommended diagnostic practice guidelines and had limited access to point-of-care diagnostic tools [22].

Self-diagnosis is equally inaccurate. In a previous study administering a questionnaire to 600 women to assess their knowledge of the symptoms and signs of VVC, only 11% of women correctly diagnosed this infection, although women who had experienced a prior episode were more often correct (35%) [23]. Self-sampling at moments of symptoms, however, can dramatically increase the sensitivity of the diagnosis of rVVC [24].

Despite the established value of microscopy in the diagnosis of VVC, cultures still seem to be the gold standard. However, cultivation of *Candida* species takes approximately 24–48 h, and even longer depending on species, which can lead to a delay in the correct diagnosis and treatment of a patient. Thus, prompt and accurate diagnosis is pivotal to achieving better outcomes. At the moment, extremely accurate and sensitive diagnostic tools are being developed, allowing physicians to diagnose VVC with substantially increased precision. Furthermore, various types of these rapid diagnostic tests have become so broadly available and affordable that self-sampling and self-testing are within reach. We aimed to summarize the current literature focusing on molecular diagnostic tools for VVC in a comprehensive review, and aim to focus on conclusive advice on proper testing today.

## 2. Materials and Methods

### 2.1. Search Strategy

A literature search was conducted using PubMed. The search was designed to retrieve studies focused on rapid diagnostic tests, specifically applied to vulvovaginal isolates of *Candida* species, focusing particularly on studies assessing and/or comparing the diagnostic parameters of molecular methods in VVC.

The search query was structured as follows:

(“vulvovaginal candidiasis” OR “vaginal candidiasis” or “vulvovaginal candidosis” OR “vaginal candidosis” OR “vaginal yeast infection” OR “*Candida* vaginitis” OR “yeast vaginitis” OR “*Candida* albicans” OR “vaginal *Candida*” OR “vaginitis” OR “vaginosis”) AND (“rapid diagnostic tests” OR “point of care testing” OR “PCR” OR “polymerase chain reaction” OR “DNA probes” OR “molecular diagnostics”) AND (“specificity” OR “sensitivity” OR “predictive value” OR “diagnostic accuracy”).

### 2.2. Inclusion and Exclusion Criteria

#### 2.2.1. Inclusion Criteria

Studies specifically addressing VVC, excluding other forms of *Candida* infections such as oral or invasive candidosis.Studies employing molecular diagnostic techniques (e.g., PCR, DNA probes) directly related to the identification and characterization of *Candida* species in vaginitis.Studies assessing the predictive value of rapid detection methods in diagnosing VVC.

#### 2.2.2. Exclusion Criteria

Articles focusing on *Candida* infections of non-vaginal sites, such as blood or systemic infections.Studies that discuss molecular diagnostics in the context of other vaginal infections that do not specifically include or differentiate *Candida* species.Articles published in languages other than English.Research involving animal models or in vitro studies without direct clinical relevance to human populations.

### 2.3. Screening and Selection Process

Articles retrieved from the initial search were first subjected to title and abstract screening to assess their relevance based on the inclusion and exclusion criteria. Additionally, we expanded our search to include relevant studies identified through reference checks of included articles and by reviewing similar articles suggested by PubMed. Subsequent full-text reviews were performed for selected studies to further evaluate their suitability for inclusion in the review.

The selection process was visualized using a flowchart which detailed the number of records identified, included, and excluded, and the reasons for exclusions during the different phases of the review (Figure 1).

### 2.4. Data Extraction and Analysis

Data from the included studies were systematically extracted using standardized tables to ensure consistent data collection and facilitate comparative analysis. The extraction process focused on study design, population characteristics, specific diagnostic methods/techniques employed, outcomes measured and/or the predictive accuracy of these methods. These data were then synthesized narratively to highlight the efficacy and practicality of various rapid diagnostic tests in clinical settings for the diagnosis of vaginal *Candida* infections.

## 3. Results

In total, 23 studies were finally included in this review [25,26,27,28,29,30,31,32,33,34,35,36,37,38,39,40,41,42,43,44,45,46,47] (Table 1), mostly assessing NAAT/PCR assays and DNA probes. Of note, both assays represent molecular techniques used to detect specific nucleic acids (DNA or RNA) in a sample, but they differ in their principles, sensitivity, and applications. Briefly, PCR/NAAT assays amplify nucleic acids to detect very small amounts of DNA/RNA with high sensitivity, have a broad spectrum of applications and require specialized equipment (thermal cycler). On the other hand, DNA probes bind directly to a target sequence without amplification, and their spectrum of application is more limited but includes gene detection and chromosome analysis, making them less sensitive but simpler in terms of detection.

Of those 23 studies above, 12 examined the diagnostic accuracy of NAAT/PCR assays [25,26,27,28,30,34,36,40,43,44,46,47], and 6 the diagnostic accuracy of DNA probes [31,32,35,38,39,42]; 3 compared PCR with DNA probes [33,41,45], and 2 referred to other assays [29,37]. Most studies came from the US or Europe and only 5 included Asian countries or Australia [25,27,29,32,37]. Of note, only one study was a randomized controlled trial [27], the majority of the rest being of observational or cross-sectional design [26,28,29,30,31,32,33,35,36,40,41,43,46,47]. In most studies, sensitivity and specificity for VVC diagnosis using molecular diagnostics was over 86% [25,26,28,29,33,34,36,43,44,45,46], way over conventional methods of diagnosis [26,27,34,43,46].

## 4. Discussion

This review aimed at exploring currently available rapid molecular diagnostic methods in VVC. A number of studies were identified including NAAT/PCR and DNA probing. We found that in most cases, molecular rapid diagnostic tests significantly outperform conventional diagnostic methods, including culture and wet mount microscopy, in terms of specificity and sensitivity in the diagnosis of VVC. However, current guidelines for the diagnosis of vaginitis include clinical evaluation, vaginal pH assessment, the “whiff” test and wet mount microscopy. Nonetheless, the current approach for the management of vaginal infections seems to provide suboptimal care [16]. In this context, molecular tests provide for rapid turnaround time, identification of NACs, and automation and standardization, avoiding human error and ensuring consistent results among laboratories. However, despite their high diagnostic capacity, their limited availability, need for technical expertise, increased cost, risk of overdiagnosis, and the lack of sensitivity data for antifungals demands caution be maintained in their use, so that rapid molecular diagnostic misuse is avoided. Vaginal candidosis should be analyzed in the context of the accompanying microbiota detected, so as to ensure if it is really candidosis, and if so, if it is a simple, complicated or recurrent case. The question of colonization without a need fortreatment, or invasive infection requiring intervention, calls for integration of different methods guided by clinical judgment, especially in settings involving mixed infections, but also in the case of increased antimicrobial resistance. 

### 4.1. NAATs/PCR Assays

Two preliminary studies have demonstrated the superior sensitivity of PCR in detecting Candida species compared to yeast cultures. PCR identified higher percentages of both symptomatic (42.3% versus 29.8% with culture) and asymptomatic patients (7.3% versus 4.9%) [47]. However, partial concordance was observed between the two methods in detecting *Candida* sp., indicating that sole reliance on one method could lead to inaccurate diagnosis of VVC [40]. NAATs have shown high sensitivity rates (92.4%) in contrast with culture (83.3%) and microscopy (48.5%), although clinical correlation is required since NAAT may identify innocent colonization with *Candida* spp. [34]. This low sensitivity for microscopy can have several reasons. One is the failure to use phase contrast microscopy as opposed to normal light transmission microscopy. Indeed, the addition of phase contrast dramatically increased the diagnostic accuracy, even amongst highly experienced experts in microscopy [48]. Also, proper training in microscopy is often lacking in currently educational programs. Still, the high-level accuracy of phase contrast microscopy can be achieved by an intensive training of only 10 h [48]. These shortcomings, amongst others, like lack of time for and lack of availability of microscopes at the location where the patient is being examined, all induce a false low performance for microscopy.

A more recent study evaluated the performance of PCR coupled with quantum dot fluorescence analysis (QDFA) for the diagnosis of *Candida* strains in leukorrhea samples from patients with suspected VVC [25]. The sensitivity and specificity of PCR-QDFA was 89.01% and 93.69% for C. albicans, 85.88% and 99.37% for *C. glabrata*, 81.25% and 99.71% for *C. tropicalis*, and 92.86% and 99.57% for *C. krusei,* respectively, suggesting that this technique can be used as a rapid (approximately 4 h) diagnostic tool for identification of the most common *Candida* strains [25]. Another study compared the conventional method of cultures to PCR for *Candida* species in women with post-antibiotic candidosis [27]. PCR was more sensitive than culture, particularly among asymptomatic women; however, whether a positive PCR result represented colonization or a true infection warranting treatment remained difficult to distinguished. PCR was not able to detect significantly more cases of VVC in symptomatic patients compared to conventional cultures, suggesting that other yeast species, rather than *Candida*, could be implicated in post-antibiotic vulvovaginitis [27].

The BD MAX Vaginal Panel for the BD MAX system is a multiplex real-time PCR-based assay for specific VVC DNA targets [38]. Several studies have compared the performance of the BD MAXTM vaginal panel with conventional methods for the diagnosis of VVC. The BD MAXTM vaginal panel had higher sensitivity and specificity in detecting Candida spp. compared to KOH preparation and clinical diagnosis [26]. No significant differences were noted between clinician-collected samples and self-swabs, reporting sensitivity of 90.9% and specificity of 94.1%, respectively [36]. Analogous rates were reported by other authors who found a sensitivity and specificity of 97.4% and 96.8% for the diagnosis of VVC [28]. In a study conducted in UK, the sensitivity and specificity of the BD MAXTM vaginal panel for all *Candida* species was 86.4% and 86.0%, respectively, indicating that this method offers benefits in settings where immediate microscopy is unavailable [44]. When the BD MAXTM vaginal panel was compared to clinical diagnosis for VVC, authors reported a positive agreement of 53.5% suggesting that the clinical diagnosis missed nearly half of the cases detected by the vaginal panel assay; however, clinical diagnosis was efficient for confirming negative results with a negative percent agreement of 77.0% (false-positive rate 23.0%) [30]. Another study compared an Aptima *Candida*/*Trichomonas* vaginitis assay with yeast cultures and DNA sequencing for VVC. Sensitivity and specificity estimates were 91.7% and 94.9% for the investigational test, with similar rates for both clinician-collected samples and patient-collected samples, suggesting that this method was more predictive of infection than traditional diagnostic methods [43]. Moreover, the performance of the Seegene Allplex™ Vaginitis assay in the diagnosis of candidiasis was recently investigated. The sensitivity and specificity of the assay test compared to yeast cultures was 91.1% and 95.6% for any *Candida* spp., 88.1% and 98.2% for *C. albicans*, and 100% and 97.5% for non-albicans *Candida*. The presence of multiple infections did not interfere with the performance of the test; nonetheless, within the subgroup of symptomatic women, a syndromic approach led to under-diagnosis (Sensitivity: 66%), and therefore we would not recommend it, even in populations at high risk for sexually transmitted diseases [46]. This observation came in line with a recent study testing a new vaginal formulation for the treatment of VVC, showing that the Seegene test for *Candida* detection had an unacceptably low sensitivity when compared to culture, microscopy, and another PCR test (unpublished results, data on file). Hence, laboratory confirmation is necessary in all settings [49].

### 4.2. DNA Probes

A DNA probe analysis of vaginal fluid provides a point-of-care option for the three common causes of acute vulvovaginal symptoms [39]. The BD Affirm VPIII microbial identification test is a multianalyte, nucleic acid probe-based assay system designed to enable the identification and differentiation of organisms associated with vaginitis [33]. The sensitivity and specificity of clinician microscopically diagnosed vulvovaginal candidiasis were 39.6% and 90.4, respectively, while the sensitivity and specificity of the DNA probe diagnosis for the same type of vaginitis were 75.0% and 95.7% [35]. Another study also documented that the Affirm assay was significantly more likely to identify *Candida* than wet mount (11% were positive by Affirm compared to 7% by wet mount). Additionally, asymptomatic women were significantly more likely to test negative by Affirm (43% versus 5% by wet mount) [31]. The detection rate achieved by the Affirm assay did not significantly differ from that of vaginal culture (13.33% versus 14.87%) with a sensitivity and specificity rate of 82.76% and 98.80% for the Affirm test compared to the diagnostic standard [32]. When compared to Pap test, Affirm VPIII was more sensitive for the detection and identification of candidiasis (16.2% by Affirm assay versus 13.9% by Pap test) [38]. In a study conducted in Greece, the sensitivity and specificity of the Affirm assay was compared to those of Gram-stain, KOH preparation, and Sabouraud culture for *Candida* spp. detection [42]. Affirm VPIII showed very satisfactory rates in both symptomatic and asymptomatic patients compared to the standard methods [42]. Of importance, several authors noted that the Affirm assay was more efficient and accurate for the diagnosis of multiple infections compared to other methods [32,35,38].

### 4.3. DNA Probes vs. PCR Tests

There are few studies comparing DNA probes (BD Affirm™ VPIII) to PCR-based methods. A recent study demonstrated that BD Max MVP detected VVC more frequently in contrast to the BD Affirm, with detection rates of 13.5% and 6%, respectively [41]. Another study reported that BD MAX vaginal panel sensitivity and specificity for detecting *Candida* spp. were 98.4% and 95.4% compared with 69.4% and 100% of the DNA probe method [45]. BD Affirm compared to CAN-PCR, a detection system that specifically detects *C. albicans* and *C. glabrata*, and showed lower sensitivity (58.1% versus 97.7%) but high specificity for *Candida* vaginitis [33].

### 4.4. Other Molecular Methods/Comparisons

Various alternative methods have been evaluated for the identification of *Candida* species and other yeast pathogens. MALDI-TOF MSs have shown high accuracy and speed for the identification of 98.3% of opportunistic yeast species and successful detection of all five top *Candida* species (*C. albicans*, *C. glabrata*, *C. parapsilosis*, *C. tropicalis*, and *P. kudriavzveii*). Even though its application requires a culture-positive sample, its use can significantly reduce time to diagnosis. API 20C AUX identified 97.26% of the most prevalent *Candida* species; however, a few clinically rare species were misidentified. The 21-plex PCR correctly identified 87.3% of all included yeast species (100% of the most prevalent *Candida* species) with a high specificity rate of 98.7% [29]. Lastly, a novel method based on gold nanoparticles has been developed and has been under investigation for rapid diagnosis of vaginal infections. In an experimental study, the method showed 100% sensitivity and specificity for the diagnosis of *Candida* vaginitis [37]

### 4.5. Study Limitations

While the reviewed studies offer significant insights into the diagnostic methods for vaginal infections, it is essential to acknowledge several limitations. Some of the included studies had small sample sizes and limited representation of certain ethnic groups. Similarly, our search was limited to English language papers and the Pubmed database, potentially missing a number of studies. Additionally, in some cases, the investigational test may have resulted in overdiagnosis of vaginitis since it could not distinguish colonization from pathogenic growth. A side-by-side comparison between the investigational test and the standard diagnostic method is not always performed, and even though extrapolation of results, while feasible, may lack precision. Lastly, although more sensitive and specific diagnostic tools have been developed, there is a notable absence of studies evaluating guidance for treatment based on these advanced methods and the impact of treatment selection on clinical outcomes [50]. To this end, future research should focus on addressing those limitations.

### 4.6. Future Perspectives

Overall, it seems that currently available and tested molecular rapid tests for the diagnosis of VVC perform quite satisfactorily. Nonetheless, the problem of concurrent presences of various infections and different types of vaginal diseases (BV, aerobic vaginitis, vaginal atrophy, or cytolytic vaginosis) persists. In this sense, tools that combine inputs such as automated microscopy, automated pH measurement, and patient-reported symptoms can enhance patient evaluation and treatment, irrespective of the caregiver’s training and skills that can be helpful [15]. Moreover, the use of artificial intelligence and neural network models in healthcare has proven to be an excellent alternative for various tasks such as risk stratification, diagnosis, prognosis, and appropriate treatment, hence an opportunity in VVC diagnosis and management [51]. These technologies enable rapid data analysis with considerably satisfactory sensitivity and specificity compared to traditional methods. Integrating such technological models with diagnostic methods has shown potential for early diagnosis and treatment in patients with VVC or candidemia [52,53].

Despite these advancements, it is important to note that molecular tests, while rapid and effective, should not yet be considered as point-of-care tests for VVC diagnosis. Alternative, cheaper, and less complex methods, including wet mount microscopy in the hands of an experienced user, could provide a definite diagnosis within minutes for a symptomatic VVC [11] and should not be abandoned without careful consideration.

## Figures and Tables

**Figure 1 diagnostics-14-02313-f001:**
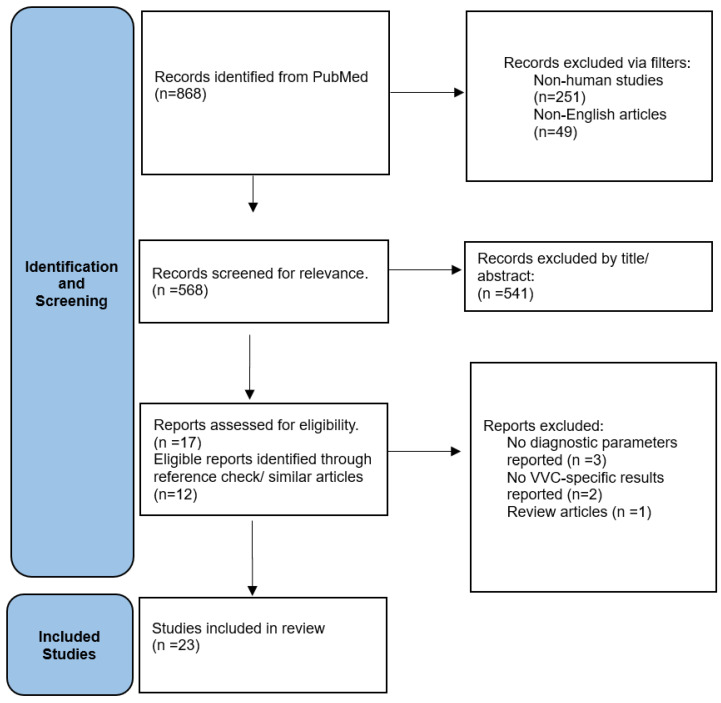
Flow Chart of study selection.

**Table 1 diagnostics-14-02313-t001:** Overview of Studies Evaluating the Diagnostic Performance of Molecular Methods for Vulvovaginal Candidosis.

Author	Year of Publication	Country	Study Design	Sample Size	Molecular Diagnostic Method Evaluated	Diagnostic Performance Metrics	Key Findings
NAATs/PCR assays							
Fan et al. [25]	2023	China	Not specified (letter to the editor)	720	PCR-QDFA	Sensitivity and specificity for *Candida* strains: *C. albicans*: 89.01%, 93.69% *C. glabrata*: 85.88%, 99.37% *C. tropicalis*: 81.25%, 99.71% *C. krusei*: 92.86%, 99.57%	PCR-QDFA is a highly sensitive and specific method that can rapidly diagnose *Candida* strains in leukorrhea samples from patients with suspected VVC.
Tabrizi et al. [27]	2006	Australia	RCT	90	PCR	PCR detected additional cases of *Candida* (4 *C. albicans*, 3 *C. parapsilosis*, 1 *C. tropicalis*) compared to culture	PCR was more sensitive than culture in detecting *Candida* in vaginal swabs post-antibiotic treatment, particularly in asymptomatic patients. PCR did not find significantly more *Candida* in symptomatic patients.
Schwebke et al. [26]	2018	USA	Diagnostic Accuracy, Cross-sectional	1613	BD MAX™ Vaginal Panel (Becton, Dickinson and Company, Franklin Lakes, NJ, USA)	BD MAX™ Vaginal Panel had higher diagnostic performance in detecting *Candida* spp. Se: 90.7%, Spe: 93.6%, OPA of 92.7%, and a κ value of 0.84. PPV: 87.2% NPV: 95.5%	The investigational test showed higher sensitivity and specificity compared to potassium hydroxide preparation and clinician diagnosis in detecting *Candida* spp. (KOH preparation: Se: 57.5% and Spe: 89.4%. Clinician Diagnosis: Se: 56.8% and Spe: 89.2%.)
Sherrard J [44]	2019	UK	Comparative Study	51 Microscopy positive culture, 19 Standard tests negative	BD MAX™ Vaginal Panel	*Candida* spp. Se: 86.4%; Spe: 86.0%	The panel offers benefits where immediate microscopy is unavailable.
Gaydos et al. [36]	2017	USA	Multicenter Cross-sectional	1618 clinician-collected, 1628 self-collected swabs.	BD MAX™ Vaginal Panel	*Candida* Group (*C. albicans*, *C. dubliniensis*, *C. parapsilosis*, and *C. tropicalis*): Se: 90.9% (95% CI 88.1–93.1%) Spe: 94.1% (95% CI 92.6–95.4%) *C. glabrata*: Se: 75.9% (95% CI 57.9–87.8%) Spe: 99.7% (95% CI 99.3–99.9%)	The investigational molecular test accurately diagnosed the common fungal causes of vaginitis with high sensitivity and specificity, applicable to both clinician and self-collected swabs.
Aguirre-Quiñonero et al. [28]	2019	Spain	Cross-sectional	1000	BD MAX™ Vaginal Panel	*Candida* spp.: Total Detected (*C. albicans*, *C. glabrata*, *C. krusei*, etc.): Se: 97.4% (95% CI, 95.1–98.6) Spe: 96.8% (95% CI, 95.2–97.9) PPV: 93.9% (95% CI, 90.9–95.9) NPV: 98.7% (95% CI, 97.5–99.3) *C. glabrata*-specific: Se and Spe: 100% (95% CI, 85.7–100) *C. krusei*-specific: Se: 83.3% (95% CI, 43.6–97.0) Spe, PPV, NPV: 100% (95% CI, 99.6–100)	Conventional methods complement molecular diagnostics, highlighting discrepancies, especially in *Candida* detection. The high sensitivity of molecular tests is probably responsible for the 6.4% false positive VVC results
Danby et al. [34]	2021	USA	Cross-sectional	300 (200 symptomatic, 100 asymptomatic)	NAAT	VVC: Microscopy Se: 48.5% Culture Se: 83.3% NAAT Sey: 92.4% Test Concordance: *Candida* species: High between culture and NAAT (91%)	NAAT shows high sensitivity and comparable or superior performance to culture in diagnosing VVC. Given the limited sensitivity of wet mount for detecting VVC, utilizing culture or NAAT is advisable when initial microscopy tests are negative.
Schwebke et al. [43]	2020	USA	Multicenter Cross-Sectional Diagnostic-Accuracy	1496 evaluated for CV in at least one sample type. (2 sample types collected: Clinician-collected and Patient-collected)	Aptima BV and Aptima *Candida*/*Trichomonas* Vaginitis (CV/TV) Assays	For clinician-collected samples, sensitivities and specificities against reference method samples: 91.7% and 94.9% for the *Candida* species group, 84.7% and 99.1% for *C. glabrata*. For patient-collected samples, sensitivities and specificities against reference method samples: 92.9% and 91.0% for the *Candida* species group, 86.2% and 98.7% for *C. glabrata*.	High diagnostic performance of Aptima assays compared to clinical diagnosis and other in-clinic assessments.
Weissenbacher et al. [47]	2009	Germany	Cohort Study	145	PCR	PCR detected *Candida* in a higher percentage of symptomatic patients (42.3%) compared to culture (29.8%). In asymptomatic patients, PCR detected *Candida* in 7.3% compared to 4.9% by culture.	PCR proved to be more sensitive than culture in identifying *Candida* species, including *C. albicans* and *C. glabrata*. PCR also provided detailed information on species prevalence, with 36.5% of symptomatic patients testing positive for *C. albicans* and 5.8% for *C. glabrata*.
Mårdh et al. [40]	2003	Sweden	Observational Study	103 sampes of women with recurrent VVC	PCR	PCR and culture showed partial concordance in detecting *Candida*. In 43.8% of cases, both PCR and culture were positive; 23.3% were negative by both. However, 20.5% were only culture-positive and 17.8% were PCR-positive only.	Discrepancies indicate that reliance solely on one method may lead to inaccurate diagnosis.
Broache et al. [30]	2021	UK	Cross-sectional Study	489	BD MAX™ Vaginal Panel Assay	Positive Percent Agreement: The clinical diagnosis agreed with the vaginal panel assay in detecting VVC in 68 cases. However, the clinical diagnosis failed to detect 59 cases that were positive by the assay, resulting in a positive percent agreement of 53.5% (95% CI: 44.5–62.4%). Negative Percent Agreement: Both the clinical diagnosis and the assay led to negative results for VVC in 261 cases. Despite this, the clinical diagnosis detected 78 cases as positive that the assay did not (false-positive rate of 23.0%), resulting in a negative percent agreement of 77.0% (95% CI: 72.1–81.4%). κ = 0.292 (0.199–0.385)	The difference in detection between the two diagnostic methods was not statistically significant (*p* = 0.124), indicating no significant discrepancy in the performance of clinical diagnosis versus the vaginal panel assay for VVC under the conditions of this study.
Vieira-Baptista et al. [46]	2021	Portugal	Cross-Sectional, Prospective Study	758	Real Time PCR (Seegene Allplex Vaginitis ™, Seegene, Seoul, Republic of Korea )	Overall Performance for *Candida* spp.: Se: 91.1% (95% CI 82.23–96.08%) Spe: 95.6% (95% CI 93.65–97.12%) PPV: 75.9% (95% CI 68.29–82.20%) NPV: 98.6% (95% CI 97.34–99.28%) κ: 0.80 (95% CI 0.735–0.864), indicating almost perfect agreement. Performance by *Candida* Species: *C. albicans*: Se: 66.7% (95% CI 29.93–92.51%) in symptomatic women, highlighting variability in detection accuracy depending on symptom presentation. Despite the lower Se in symptomatic cases, there was still substantial agreement with the reference standard (κ = 0.73, 95% CI 0.471–0.981). NAC: Se: 100% (95% CI 85.18–100.00%), indicating excellent detection capability. Spe: 97.5% (95% CI 95.73–98.38%), showing high accuracy in correctly identifying NAC infections.	The molecular assay outperformed traditional methods like wet mount microscopy and Amsel criteria, especially in detecting multiple infections and NAC, with NAC Se reaching 100%.
PCR vs. DNA probe							
Navarathna et al. [41]	2023	USA	Retrospective Analysis	8878 orders placed for DNA probe–based identification (ID) and 10,464 total orders placed for molecular panel ID.	BD Affirm™ VPIII DNA probe method and BD MAX™ MVP (Molecular Panel ID (Becton, Dickinson and Company, Franklin Lakes, NJ, USA))	VVC Detection: BD MAX™ MVP reported a higher detection rate (13.5%) for *Candida* species compared to BD Affirm™ (6%) The proportion of tests that were positive for the BD MAX™ MVP was 0.073 (0.065–0.081) higher than for BD Affirm™	Age differences are noted in *Candida* infections, with older patients more likely to have *C. glabrata*. [6.7 years older on average (range 5.2–8.1 years)] The study also examined clinical decision-making, revealing that many cases of *N. glabrata* and *Pichia kudriavzevii* (formerly *C. krusei*) were not treated following current CDC guidelines
Cartwright et al. [33]	2013	USA	Observational Study	323	CAN-PCR ((Luminex Inc., Madison, WI)), BD Affirm™ VPIII (Becton, Dickinson, Sparks, MD, USA)	CAN-PCR demonstrated Se: 97.7% and Spe: 93.2% for detecting *Candida* species. Affirm VPIII showed Se: 58.1% but Spe: 100% for Candida vaginitis.	Both tests were assessed in a population primarily composed of African American women with clinically documented vaginitis syndrome.
Thompson et al. [45]	2020	USA	Comparative Study	200	BD Affirm™ VPIII (Becton, Dickinson and Company, Sparks, MD) and BD MAX™ Vaginal Panel ((MAX VP; Becton, Dickinson and Company, Quebec, Canada))	For *Candida* spp.: MAX VP Se: 98.4% and Spe: 95.4%; Affirm showed Se:69.4% and Spe:100%	For MAX VP- *Candida*, 88.9% (56/63) of the detected *Candida* cases, including 55 *C. albicans* and 1 *C. dubliniensis*, were confirmed by culture, achieving an overall agreement with culture of 95.9%. (98.3% positive agreement and 94% negative agreement). Additionally, for *C. glabrata*, 88.9% of samples identified as positive by MAX VP were confirmed by culture, resulting in an overall culture agreement of 99.5%, (100% positive agreement and 99.5% negative agreement). In comparison, the standard-of-care Affirm test identified eight true-positive cases of *C. glabrata*, all of which were confirmed by culture and the MAX VP.
DNA probes							
Byun et al. [32]	2016	Republic of Korea	Observational Study	195	Affirm VPIII Microbial Identification Test	The detection rates achieved by each detection method (Affirm assay vs. diagnostic standard) for *Candida* were not significantly different (13.33% vs. 14.87% for *Candida*). For *Candida*, the sensitivity and specificity of the Affirm test compared to the diagnostic standard were 82.76% and 98.80%, respectively	Overall, the Affirm test had an absolute accuracy of 79.5% in detecting BV, VVC, and TV, with the highest accuracy for negative results (87.9%) and mixed infections (60%).
Brown et al. [31]	2004	USA	Cohort Study	425	The Affirm VPIII test (Becton Dickinson and Company, Sparks, MD, USA)	Affirm detected higher rates of *Candida* (11% vs. 7%) compared to wet mount	Symptomatic women were significantly more likely to test positive by Affirm only (23% vs. 10% in asymptomatic women), by wet mount only (3% vs. 2%), and by both Affirm and wet mount (15% vs. 1%). Conversely, 43% of asymptomatic women were negative for both tests compared to only 5% of symptomatic women.
Levi et al. [38]	2011	USA	Comparative Study	431	Affirm VPIII molecular test (Becton Dickinson, Burlington, NC, UCA)	Detection Rates: Pap Test: Detected *Candida* in 60 patients, which constituted 13.9% of the total cases. Affirm VPIII Assay: Detected *Candida* in 70 patients, corresponding to 16.2% of the cases. The agreement between the Pap test and the Affirm VPIII assay for the diagnosis of Candida infection was quantified using the kappa (κ) statistic, which was calculated to be 0.66 (good agreement)	The Affirm VPIII detected higher rates of *Gardnerella, Candida*, and *Trichomonas* compared to the Pap test. The κ value indicated good agreement for *Candida* diagnosis (0.66), but poor agreement for BV (0.32) and *Trichomonas* (0.30). Many cases had co-infections, which were more frequently detected by the Affirm VPIII.
Lowe et al. [39]	2009	USA	Prospective Comparative Study	535	BD Affirm™ VPIII microbial identification system. BD Diagnostics; Sparks (MD)	Clinical diagnosis results compared to DNA probe: Sensitivity 83.8%, Specificity 84.8% PPV: 67.8% (60.4–74.4) NPV: 93.2% (89.9–95.5)	
Ferris et al. [35]	1995	USA	Comparative Study	501	DNA Hybridization Test	Clinician-diagnosed VVC Se: 39.6% and Spe:90.4%, DNA probe diagnosis Se:75.0% and Spe:95.7%	Clinical microscopy showed lower Se but high Spe in diagnosing vulvovaginal candidiasis compared to DNA hybridization tests. Clinicians struggled with the diagnosis of multiple infections.
Petrikkos et al. [42]	2007	Greece	Comparative Study	291	ffirm VP III Microbial Identification Test (Becton Dickinson Microbiology Systems,	Sensitivity: 81.8% (Affirm vs. KOH), 75% (Affirm vs. Gram), 60% (Affirm vs. Sabouraud) Specificity: 99.6% (Affirm vs. KOH), 93.9% (Affirm vs. Gram), 99.6% (Affirm vs. Sabouraud)	The test was efficient for both symptomatic and asymptomatic women, providing results quickly (30–45 min) and safely.
Other methods/Comparisons							
Arastehfar. [29]	2019	Iran, China, The Netherlands	Retrospective Analysis	301	MALDI-TOF MS, 21-plex PCR, API 20C AUX, LSU rDNA Sequencing (reference)	MALDI-TOF MS: Identified 98.33% of the isolates correctly. 21-plex PCR: Correctly identified 88.7% of the isolates. API 20C AUX: Correctly identified 83.7% of the isolates. κ: Indicated high agreement between the methods and sequencing, with MALDI-TOF MS at 0.991, 21-plex PCR at 0.943, and API 20C AUX at 0.918.	MALDI-TOF MS was the most effective method in this study for identifying yeast isolates. The 21-plex PCR also showed high efficacy, especially for the more prevalent *Candida* species. Integration of PCR and API 20C AUX could optimize yeast species identification, combining the rapidity of PCR and the comprehensiveness of API 20C AUX.
Hashemi et al. [37]	2019	Iran	Experimental Study	635	Gold Nanoparticle-Based Agglutination Test	For *Candida* spp., both Se and Spe were 100%.	The method efficiently differentiated between the pathogens (*Gardnerella, Candida, Trichomonas*) with high predictive values.

NAAT: Nucleic Acid Amplification Test; PCR: polymerase chain reaction; VVC: Vulvovaginal Candidosis; QDFA: quadratic discriminant function analysis; PPV: Positive Predictive Value; NPV: Negative Predictive Value; Se: sensitivity; Spe: specificity; MALDI-TOF: matrix-assisted laser desorption/ionization—Time Of Flight; OPA: Overall Percent Agreement; RCT: Randomized Clinical Trial; NAC: non-albicans candida; BV: bacterial vaginosis; TV: Trichomonas Vaginosis.

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
