# Peer review of "Rapid Molecular Diagnostics in Vulvovaginal Candidosis"

_diagnostics, 2024, doi:10.3390/diagnostics14202313_

Round 1
Reviewer 1 Report
Comments and Suggestions for Authors
Dear authors:
I have reviewed your work; it seems to me to be correct in terms of information, arrangement and wording.
Minor suggestions have been added to the pdf file, as well as a question mark.
thank you very much

Author Response
Dear authors:
I have reviewed your work; it seems to me to be correct in terms of information, arrangement and wording.
Minor suggestions have been added to the pdf file, as well as a question mark.
thank you very much
Thank you for your input. Now all your comments have been addressed and all your suggestions adopted
Reviewer 2 Report
Comments and Suggestions for Authors
Dear authors: Here I point out some considerations to your article
1. Sometimes in the text Candida is not written in italics, please correct.
2. I believe that identification tests employing Maldi-toff are very useful,but isolation is required beforehand. These tests are not for diagnosis of CVV, but for subsequent identification of yeasts. I do not understand why you mention those articles here.
3. In the introduction you say, “Compared to the clinical diagnosis, the microbiological diagnosis exhibits a sensitivity of 70.3% and a specificity of 83,7%” This statement was taken from a Nigerian study, so is it legitimate to generalize these data to the rest of the world?
4. Candida is part of the normal vaginal microbiota, as you rightly point out, and its presence in small quantities, even when isolated in cultures, does not indicate that it is the causative agent of vaginosis. For diagnosis it is really important to evidence its presence in wet mount or Gram preparations of vaginal discharge samples. On the other hand, the diagnosis of vaginal candidiasis should be analyzed in the context of the accompanying microbiota to know if it is really candidiasis and then if it is a simple, complicated or recurrent candidiasis. Treatment differs according to the whole picture.
5. I consider this is a good article about the efficacy of different molecular methods to evidence the presence of yeasts, but these methodologies do not allow to differentiate between those cases where Candida is responsible for the pathology and those where it is only a part of normal microbiota.
6. Other considerations are cost-effectiveness of these molecular techniques vs. a direct examination of a vaginal sample by a wet preparation and a Gram stain that which is fast, economical and allows the evaluation of all the microorganisms present in order to reach the correct diagnosis.
Author Response

(The authors gave the same response as above.)

Round 2
Reviewer 2 Report
Comments and Suggestions for Authors
Dear authors
The authors have made some modifications, and the discussion and conclusions explain the limitations and problems of an excessive use of these molecular techniques as to base the diagnosis of vulvovaginitis solely on their use.
Regarding the use of proteomic techniques such as Maldi-Toff in the correct identification of species, this is very useful and provides results with a high level of certainty and very quickly. However, this data is not a necessary element in the initial diagnosis of vulvovaginal candidiasis.